# Automatic Detection of Chewing and Swallowing [note 1]

**DOI:** 10.3390/s21103378

**Published:** 2021-05-12

**Authors:** Akihiro Nakamura, Takato Saito, Daizo Ikeda, Ken Ohta, Hiroshi Mineno, Masafumi Nishimura

**Affiliations:** 1Graduate School of Integrated Science and Technology, Shizuoka University, Hamamatsu 432-8011, Japan; mineno@inf.shizuoka.ac.jp (H.M.); nisimura@inf.shizuoka.ac.jp (M.N.); 2NTT DOCOMO, Inc., Tokyo 100-6150, Japan; takato.saitou.bu@nttdocomo.com (T.S.); ikedad@nttdocomo.com (D.I.); ootaken@nttdocomo.com (K.O.)

**Keywords:** chewing, swallowing, eating behavior, hybrid CTC/attention model, data augmentation

## Abstract

A series of eating behaviors, including chewing and swallowing, is considered to be crucial to the maintenance of good health. However, most such behaviors occur within the human body, and highly invasive methods such as X-rays and fiberscopes must be utilized to collect accurate behavioral data. A simpler method of measurement is needed in healthcare and medical fields; hence, the present study concerns the development of a method to automatically recognize a series of eating behaviors from the sounds produced during eating. The automatic detection of left chewing, right chewing, front biting, and swallowing was tested through the deployment of the hybrid CTC/attention model, which uses sound recorded through 2ch microphones under the ear and weak labeled data as training data to detect the balance of chewing and swallowing. N-gram based data augmentation was first performed using weak labeled data to generate many weak labeled eating sounds to augment the training data. The detection performance was improved through the use of the hybrid CTC/attention model, which can learn the context. In addition, the study confirmed a similar detection performance for open and closed foods.

## 1. Introduction

A series of eating behaviors, including chewing and swallowing, are considered important from the viewpoint of health maintenance. It has been reported that people who chew less per swallow tend to eat more quickly, leading to obesity and lifestyle-related diseases [1,2]. Moreover, partial chewing causes tooth loss and facial distortion. Healthcare and medical fields require methods of accurately monitoring oral behavior.

Various methods are currently being investigated to collect eating behavior data. Methods using fiberscopes and X-rays have been proposed. For example, one study analyzed the relationship between the number of chewing cycles and the grinding state of the bolus using a fiberscope [3]. Similarly, investigations have used fiberscopes to observe food masses to evaluate swallowing functions [4]. Such methods can be used to detect swallowing; however, no extant method can accurately capture the state of chewing. In addition, the use of fiberscopes is highly invasive because it mandates radiation exposure during data collection.

Some less invasive methods have been suggested. Techniques using inertial measurement units (IMU), piezoelectric sensors, accelerometers, and other equipment do exist. Fontana et al. [5] detected food intake using an IMU and an accelerometer; Nguyen et al. [6] utilized long short-term memory (LSTM) to estimate the number of swallows using a piezoelectric sensor and IMU around the neck. Bedri et al. [7] recorded eating sounds using IMU and microphones and estimated chewing by setting an appropriate threshold. Farooq et al. [8] trained DNN using piezoelectric sensors and tried to detect the number of chewing instances. IMU and piezoelectric sensors are not suitable for daily monitoring because of their cost and the physical burden of wearing them over a long time. In addition, piezoelectric sensors and accelerometers evince low detection accuracy, and it is often difficult to obtain precise data for the analysis of eating behaviors.

The use of images and videos to recognize eating behaviors has also been posited: one study detected swallowing through video and machine training [9]. Such methods are less invasive for the subjects and can be easily collected; however, it is difficult to casually collect a large amount of data through this technique because of the reduced anonymity and the need to protect the privacy of subjects.

Since the measurement methods mentioned above are costly and thus unsuitable for daily monitoring, a simpler measurement technique is warranted. Sound information is characterized by a degree of anonymity in comparison to images and videos and can be collected more easily than methods using other devices. Sound recording requires only the attachment of a simple microphone and confronts fewer privacy issues than video. In addition, sound contains much information and can be used to analyze eating behavior more accurately. Yin et al. [10] attempted to estimate the ingestion of food and beverages using sound information obtained from a throat microphone and reported that they could infer whether solid or liquid foods were being consumed. In addition, Olubanjo et al. [11] estimated swallowing using a throat microphone and setting thresholds for peak frequency and other parameters.

These methods use subject information to set thresholds or heuristic approaches and are not guaranteed to precisely project subjects or foods. The present study postulated a method of training a general-purpose machine learning model using eating sounds to detect eating behaviors. Recurrent neural networks (RNNs) such as LSTM have also been investigated for the detection of eating behaviors. Ando et al. [12] attempted to identify chewing, swallowing, and speech in frames using Gaussian Mixture Model and tried to estimate them by combining them with LSTM using throat microphone recordings. However, the training data needed accurately labeling (strong label) for each frame, and it was thus impossible to obtain satisfactory performance using a small amount of data. On the other hand, Billah et al. [13] demonstrated that LSTM-Connectionist Temporal Classification (CTC) could significantly improve the recognition of chewing and swallowing by utilizing a large amount of training data with only weak labels and without accurate temporal information.

Nevertheless, the cited studies are limited to the simple detection of the number of times and approximation of eating behaviors and have not yet been able to attain information that can be used to analyze eating behaviors. Therefore, the current study examined the automatic detection of left chewing, right chewing, front biting, and swallowing using a hybrid CTC/attention model with 2ch microphones under the ear and weak labeled data for training to detect the balance of chewing and swallowing. N-gram-based data augmentation was initially performed using weak labeled data to generate a large number of weak labeled eating sounds to augment the training data. The detection performance was improved using the hybrid CTC/attention model, which can learn the context. It was also confirmed that the detection performance was similar for open and closed foods.

## 2. Materials and Methods

### 2.1. Hybrid CTC/Attention Model

#### 2.1.1. End-to-End Speech Recognition

Automatic Speech Recognition (ASR) denotes the task of inferring word sequences corresponding to input speech. Conventional ASR systems usually comprise numerous modules such as a pronunciation dictionary, an acoustic model, a language model, and a decoder.

The recent advances in computer performance and the consequent progress in research initiatives have led to studies being conducted to represent a speech recognition model encompassing multiple modules via a single neural network using deep training. This model is named “end-to-end speech recognition,” and this framework allows the entire speech recognition model to be optimized. To do so, the network is trained using a single system that inputs acoustic features derived from speech and voice and outputs phoneme sequences, words, and sentences. No explicit expert knowledge of pronunciation dictionaries or acoustic models is required, and the structure is simple because the network is represented by a single system. End-to-end speech recognition requires a large amount of data; however, the extant reports suggest that speech recognition can be performed with more accuracy than conventional models because the process does not require the adoption of a heuristic approach.

RNN networks such as LSTM that can handle time-series data are commonly utilized for end-to-end speech recognition. There are two approaches to end-to-end speech recognition using RNNs: CTC loss, and the training of an encoder-decoder model using the attention mechanism.

The state-of-art method in the ASR task is based on the transformer model, which is an extension of the encoder–decoder model using the attention mechanism. The conformer model uses CNN in combination with the transformer, which is the current state-of-art method. It achieves word error rates of 1.7%/3.3% for the LibriSpeech task [14].

#### 2.1.2. Connectionist Temporal Classification (CTC)

CTC is a loss function developed in the domain of speech recognition to build an end-to-end speech recognition system that outputs words and sentences directly from sound features [15]. In CTC, a blank label called blank is introduced between symbols to allow each symbol to be consecutively output. The input and output lengths are matched for consistency, enabling training with weak labels. Researchers can thus absorb the differences between the input and output series books and estimate the alignment between them without using Hidden Markov Models. 

Let $x=\left\{x^{1},x^{2},\cdots ,x^{T}\right\}$ be the input data sequence, $y=\left\{y^{1},y^{2},\cdots ,y^{T}\right\}$ be the output of the network, and $l=\left\{l^{1},l^{2},\cdots ,l^{T^{\prime }}\right\}$ be the correct label sequence. Generally, the length $T^{\prime }$ of l is shorter than the length T of the input data sequence. A redundant label sequence $\pi =\left\{\pi^{1},\pi^{2},\cdots ,\pi^{T}\right\}$ corresponding to l and a function B are defined to map π to l. Function B removes the same sequence of labels as “blank” and returns the final phoneme sequence. Then, $p\left(\pi |x\right)$ is expressed by (1).
(1)$$\left(\pi |x\right)=\overset{T}{\underset{t=1}{\prod }}y_{\pi_{t}}^{t}.$$

The probability $p\left(l|x\right)$ that the input/output sequence is the label sequence l is expressed by (2) using $p\left(\pi |x\right)$.
(2)$$p\left(l|x\right)=\underset{\pi \in B^{-1}\left(l\right)}{\sum }p\left(\pi |x\right).$$

The parameter that maximizes $p\left(l|x\right)$ is obtained using the maximum likelihood estimation, and the training model is created. The output label can be found when decoding by generating the label that offers the highest probability of occurrence at each time. Beam search is used to achieve this task.

#### 2.1.3. Encoder-Decoder Model Using Attention

The attention mechanism is a framework introduced in the encoder-decoder model. It has been applied to a wide range of tasks such as neural machine translation [16] and speech recognition [17]. The encoder takes input and generates a fixed-length vector h in LSTM. In the decoder, the output $y=\left\{y_{1},y_{2},\cdots ,y_{T^{\prime }}\right\}$ is generated via LSTM from the vector generated by the encoder. The t-th hidden state in the decoder $q_{t}$ is computed by (3) using the output $y_{t-1}$ of the previous decoder and the hidden layer state $q_{t-1}$ of the previous RNN.
(3)$$q_{t}=\mathrm{RNN}\left(q_{t-1},y_{t-1}\right).$$

The encoder-decoder model can be trained efficiently for short time-series data. However, it is impossible to place all the information into a fixed-length context vector for long time-series data. Thus, enough information cannot be conveyed to the decoder.

The attention mechanism is, therefore, introduced; in addition to the previous out-put $y_{t-1}$ and the state of the hidden layer $q_{t-1}$, the structure is changed so that the hid-den layer of the decoder can be determined by referring directly to the state of the encoder at each time. In this instance, the weighting is such that the state most relevant to the current time is selected. Specifically, it is expressed by (4) and (5).
(4)$$q_{t}=\mathrm{RNN}\left(q_{t-1},y_{t-1},c_{t}\right)$$
(5)$$r_{t}=\overset{T}{\underset{i=1}{\sum }}\alpha_{t,i}h_{i},$$
where $\alpha_{t,i}$ is the weight of the attention at time t, which signifies the rate of utilization of the hidden layer state $h_{i}$ at each time in the encoder. In contrast to the CTC, which assumes that each event occurs independently, attention makes it possible to reflect the history of past outputs and learn a set of contexts.

#### 2.1.4. Hybrid CTC/Attention Model

Unlike the method using CTC, the technique that utilizes attention does not assume the independence of each event in the output, which enables the learning of the context. However, words may be replaced in the output in language processing tasks such as machine translation, while speech recognition is different because the time-series relationship between input and output is maintained. It is known that the training technique converges faster than the method using CTC; hence, there are advantages and disadvantages in the use of CTC and attention.

The final output of the hybrid CTC/attention model is the sum of the output vectors of CTC and attention, which offers the benefit of taking advantage of the characteristics of both CTC and attention. The loss function L during training is defined as the weighted linear sum of the loss functions of CTC and attention and is expressed by (6)
(6)$$L=\alpha L_{CTC}+\left(1-\alpha \right)L_{Attention},$$
here, α is a hyperparameter that is trained by setting an appropriate value.

The hybrid CTC/attention model attracts attention because it uses CTC, which results in fast convergence of training, and can also learn contexts through attention.

Watanabe et al. [18] evinced that a hybrid CTC/attention model with both CTC and attention for speech recognition results in a significant improvement in precision. They reported through an experiment that used the Corpus of Spontaneous Japanese (CSJ) in which, compared to character error rate of 9.4% using only CTC and 11.4% using only attention, the character error rate (CER) improved to 8.4% when the hybrid model was utilized.

### 2.2. Automatic Detection System

Figure 1 illustrates the flow of the automatic detection system. The system classifies the eating sounds recorded by the microphones under the ear into five categories: left chewing, right chewing, front biting and swallowing, swallowing, and others. The category of “others” includes silence and noise.

#### 2.2.1. Recording of Eating Sounds

First, eating sounds are recorded by two microphones (16 bit, 22 kHz sampling) placed under the ear. Figure 2 displays an example of how the microphone is attached and also presents an enlarged image of the microphone unit. The microphones were constructed in-house by the researchers on a 3D printer.

Weak labels do not have accurate time information. An online application to assign labels was employed to reduce the cost of labeling, and weak labels were assigned to each event to create a training model. Figure 3 shows an example of the recorded eating sounds and their corresponding logs.

#### 2.2.2. Feature Extraction

The next step involved the conversion of the recorded eating sounds into speech features extracted in the window width of 80 ms and the frameshift of 40 ms. In addition to the 39-dimensional Mel-frequency cepstral coefficients (MFCC) extracted by adding the left and right signals for signal enhancement, the feature value was obtained by adding the seven-dimensional cross-correlation value to improve the detection performance for chewing position. The MFCC comprised 12 units with one-dimensional root mean square (RMS), 13-dimensional Δ, and 13-dimensional ΔΔ.

#### 2.2.3. Hybrid CTC/Attention Model

The obtained acoustic features were input to the hybrid CTC/attention model, an automatic detector, to estimate the left chewing, right chewing, front biting, and swallowing. As has been noted above, the hybrid CTC/attention model was applied to the automatic detection of eating behavior because, in comparison to conventional methods, it has demonstrated higher detection performance in speech recognition.

Attention can reflect the history of past complex eating behavior. Therefore, besides the detection of chewing and swallowing events via CTC, attention is expected to improve the accuracy of event detection. Figure 4 and Figure 5 present a sample event sequence of one cracker being eaten (Ritz). First, front biting occurs (see Figure 5a), then left and right chewing are performed in succession to some extent (see Figure 5b), and then swallowing takes place (see Figure 5c). Therefore, the sequence—including the chewing position and swallowing—may be denoted as an event series with a context.

#### 2.2.4. Double Threshold Method

Double threshold [19] was applied in this study as a smoothing process during the application of the SoftMax function in the network output so that the alternation of chewing was not generated in a short duration. This method uses two thresholds, namely, $\phi_{low}$ and $\phi_{high}$. First, it sweeps the column of estimated output probabilities and marks all elements above $\phi_{high}$ as a valid prediction frame. Second, for each valid prediction flame, it expands the marked frame by searching for continuous predictions above $\phi_{low}$. This process had the effect of correcting chewing on one side when chewing changes occurred in a short duration.

### 2.3. Data Augmentation

An automatic detection system was first created using weak labeled eating sounds for training and CTC, as depicted in Figure 6. Next, these eating sounds were input to the automatic detection system, and CTC was used to estimate the alignment of each chewing and swallowing event. The sound data corresponding to the events detected by the estimation were collected to create a database that was used for data augmentation., The interval between the start of the corresponding alignment and the beginning of the next alignment was extracted for data pertaining to each sound.

Next, as shown in Figure 7, the training data were augmented using the eating event sound data collected by the alignment estimation and the 5-gram model. The 5-gram model was constructed using the chewing (left/right) and biting (front) labels from the weak labels included in the training data. The next type of chewing (left/right) and biting (front) was determined by generating random numbers with probability based on the 5-gram model, and the sequence of events was repeatedly generated. However, a normal distribution was assumed for swallowing apropos the distribution of the number of chewing instances. The average number of times chewing occurred from the beginning of the eating to the swallowing in the training data μ, and the standard deviation σ, were calculated to obtain a normally distributed random number. This number was then used as the sum of the times a series of chewing events was executed until the event occurred.

Finally, Figure 8 evinces that eating sounds were generated by selecting and concatenating sound data corresponding to respective events of the generated event sequence from candidates. For the waveform to be used in this case, sound data representing a similar position of the head should be selected as much as possible. Let n be the number of candidate event sound data to be concatenated, and $t_{i}$ be the position from the head of the $i\;\left(1\;\leq \;i\;\leq \;n\right)$-th eating event sound data. The position from the beginning of the eating sound currently being generated is defined as c. Of all the candidates, the i-th eating event sound data was randomly selected at a rate of $\frac{1}{\left|t_{i}-c\right|+1}$. The difference of the chewing sound by the change of the shape of the food by the chewing was reflected by selecting the sound data in which the time was close. The selection of sound data close to the current time enabled the maintenance of the feature that the chewing sound in the first half before the food was crushed was higher in volume and the chewing sound in the second half was lower.

## 3. Experimental Setup

### 3.1. Dataset

The dataset included 2-channel eating sounds recorded via skin-contacting left and right microphones placed under the ear.

The food sounds of chewing gum, crackers (Ritz), and cabbage (shredded) were recorded for 26 men and four women in their 20s for the training and evaluation data for closed food. Foods with different textures were selected. Each subject ate chewing gum for 3 min, chewed a piece of cracker 26 times, and chewed 7 g of cabbage 20 times. At the same time, weak labels were assigned based on the subject’s self-report, and data were collected for 13,713 instances of left chewing, 13,269 instances of right chewing, 2853 cases of front biting, and 3308 events of swallowing.

The eating sounds of apples and pizzas for five subjects in their 20s (four men and one woman), who were not used for the training data, were recorded for the evaluation data for open foods. Weak labeled data were collected from 380 instances for left chewing, 384 cases of right chewing, and 40 swallowing events.

The correct strong labels were assigned to each frame unit of all eating sounds for evaluation. This study used the self-reported weak labeled data as a base, and the sound waveform and video recorded at the same time as the weak labeled data were utilized as references. Strong labels were assigned for frame-by-frame evaluation and were not at all employed in the training data.

### 3.2. Experimental Conditions

Several experiments were set up to evaluate the proposed method.

The first experiment compared the models and features that were utilized. The performance of models trained with LSTM using CTC as the loss function was compared to the results yielded by the encoder-decoder model using the attention mechanism as an appraisal of the proposed method that deployed the hybrid CTC/attention model. In the hybrid CTC/attention model, α indicates the ratio of the weighted sum of the loss functions of CTC and attention; in the present study, α was set to 0.7, 0.0, and 1.0. The model that only uses CTC does not consider context, and the model that only employs attention is not trained using the property that the input and output columns correspond in chronological order.

The present study experimented with the MFCCs of the sum of left and right signals and cross-correlation alone to compare features. It also tested the changing of the dimensionality of the cross-correlation in addition to the proposed concatenation of MFCCs of the sum of left and right signals and 7-dimensional cross-correlation. Cross-validation was performed by dividing the training data of 30 subjects into six parts to make the experiment open subject, and the average of each result was used as the final detection performance. Real eating sounds were used with weak labeled correct answers assigned to around 26,000 chewing positions to generate the training model. The evaluation was accomplished using around 6000 real eating sounds, including chewing and swallowing, of speakers not used for training of the remaining speakers and their corresponding correct strong labels. The correct strong labels were not used for training at all and were only employed for evaluation purposes.

The second experiment involved a comparison of data augmentation, in which the performance of the proposed data augmentation was evaluated vis-à-vis the training data of the hybrid CTC/attention model. The performance of the proposed data augmentation was compared to the training data of the hybrid CTC/attention model using only the real eating sound data and the data augmentation of 1, 3, 5, 10, and 15 instances of the real eating sound data. In addition, two aspects of the performance of the n-gram model used to generate the event sequence of chewing and swallowing during data augmentation were compared: when the event sequence was generated randomly, and when three types of models (2-gram, 5-gram, and 7-gram) were used. About 30,000 real eating sounds (including chewing and swallowing) with weak correct labels and eating sounds of all foods for all speakers generated through data augmentation were used to generate the training model. Data augmentation was performed for each speaker for each food using all real eating sounds and weak labels. No threshold was set for the number of swallows for chewing gum because no swallowing occurs. The training and evaluation data were the same as the data utilized in the first experiment.

The third experiment evaluated the detection performance using apples and pizza, foods that were different from those used for the training data. The hybrid CTC/attention model was trained with 10 times more data augmentation than the original data. In addition, eating sounds were recorded from subjects who were not included in the training data to ensure the open subject condition. The training data were the same as those used for the first experimental setup, and the evaluation data comprised an open set of foods containing around 1000 real eating sounds, including chewing and swallowing and their corresponding strong labels.

### 3.3. Metric

The mean absolute percentage error (MAPE) estimated on a frame-by-frame basis was applied as an evaluation index of the overall detection performance. Here, N denoted the number of pieces of data, $A_{k}$ indicated the number of correct frames, and $F_{k}$ represented the number of estimated frames calculated according to (7).
(7)$$\mathrm{MAPE}=\frac{100}{N}\overset{N}{\underset{k=1}{\sum }}\left|\frac{A_{k}-F_{k}}{A_{k}}\right|.$$

In addition, recall, precision, and F1 scores were also utilized for each event as the evaluation index values of detection performance for each event. These were calculated according to (8), (9), and (10), respectively.
(8)$$Recall=\frac{True\;Positives}{True\;Positives+False\;Negatives}$$
(9)$$Precsion=\frac{True\;Positives}{True\;Positives+False\;Positives}$$
(10)$$F1\;score=2\times \frac{Precision\times Recall}{Precision+Recall}.$$

The system deemed that accurate detection had occurred when it identified one or more points of overlap between the correct and estimated label.

## 4. Experimental Results and Discussion

### 4.1. Performance Comparison by Model and Sound Feature

Table 1 displays the comparison of the overall event detection performance of each model. Five categories of events were included: left chewing, right chewing, front biting, swallowing, and others. The hybrid CTC/attention model returned a much superior performance in comparison to the models that only used attention or CTC. These results suggest that rather than the speech recognition task, the consideration of the context by training with the attention mechanism and the estimation of events in chronological order by training with CTC were both effective for the task of the automatic recognition of eating behavior. The use of CTC-enabled training with easily collectable weak labeled data and was shown to be effective considering the context of eating behavior. Furthermore, the results indicated that it is possible to achieve a considerably high level of eating behavior recognition using only non-invasive sound information.

Table 2 exhibits the detection performance for each event. The use of the hybrid CTC/attention model improved the detection performance of each event. In particular, the use of this model yielded superior accuracy for left and right chewing. The detection performance of front biting and swallowing was found to be lower than the identification of left and right chewing. However, this outcome may have occurred because the recording at the left and right position below the ear was specialized for the detection of left and right chewing and the amount of training data was insufficient compared to left and right chewing. It is therefore believed that detection performance could be improved through the use of an additional microphone and more data augmentation.

Table 3 overviews a comparison of the detection performance of the entire event for each feature. The detection performance was low when only the left or the right signal was used because the information of left and right chewing was barely included. The proposed feature that extracted the MFCC of the sum of the left and right signals and conducted a 7-dimensional cross-correlation of left and right signals demonstrated the highest detection performance. It is thus contemplated that high detection performance can be ensured when a sufficient amount of training data exist, without explicitly training by extracting cross-correlations. However, it is better to extract the proposed features when the training data are limited in number, as in the present instance.

### 4.2. Comparison of Performance with and without Data Augmentation

A comparison of the detection performance achieved with and without data augmentation when the n-gram model to be trained during data augmentation was varied is shown in Table 4. As can be noted, the 5-gram and 7-gram models improved the detection performance, indicating the impact of data augmentation. Conversely, the detection performance decreased in comparison to the instance of identification without data augmentation when the event sequence was randomly generated or when a 2-gram model was applied. It is posited that the event sequences generated by these models do not reflect the context of the sequences present in real eating sounds that comprise chewing positions and swallowing. For example, the feature that left and right chewing appear in certain consecutive chunks is not sufficiently maintained when event sequences are randomly generated or when a 2-gram model is used. Therefore, such sequences may exert a negative influence on the training of context in the attention mechanism.

Figure 9 evinces the amount of data augmentation and the overall detection performance. The overall detection performance improved when data augmentation was performed than the instance without data augmentation. A ceiling of between 5 and 10 times can also be noted in the detection performance, suggesting that such amounts of data augmentation are appropriate for context training.

Table 5 shows the detection performance for each event. The accuracy of each model improved through data augmentation, and the detection performance ameliorated for all events. In particular, the identification of the less frequent events of front biting and swallowing evinced superior results through data augmentation, indicating that the training of the context in attention was advanced through the use of the large amount of data created via data augmentation.

Table 6 presents the detection performance of each food, evaluated simply through the detection of left and right chewing positions. Almost identical detection performances were obtained for all food items, including open foods. In addition, higher accuracy was registered in the detection of chewing of 3 classes than the identification of the chewing position of 5 classes. This outcome indicates that in some instances, chewing was detected, but the identification of the chewing position was incorrect.

The augmentation method that takes into account the data time series was considered, and it was not so effective. Thus, in the future, we should combine spectral transformations, such as Spec Augment Method [20] and mixup [21], that have been effective in other fields.

### 4.3. Verification of Practicality

The detection performance was evaluated using apples and pizza, food items different from foods used for the training data, to verify the practicality of the model. The hybrid CTC/attention model was trained with 10 times more data augmentation than the original data. Table 7 displays the detection performance of left and right chewing for each food. It was found that the detection performance of open foods was almost identical to the identification of closed food. The results also revealed that the chewing position could, in general, be automatically detected for food not included in the training data. Although a method for robust and efficient detection of eating behavior for common food has not been established, it was confirmed that the method can detect with high accuracy, including common food. This suggests that our method has a high degree of practicality.

Figure 10 outlines the relationships between the detection performance of each food item and the time elapsed from the beginning of eating. The results for closed foods such as crackers, chewing gum, and cabbage are also shown for comparison. The time-lapse from the start of eating was normalized by dividing the duration between the food being placed in the mouth to its being swallowed into five equal parts. The chewing position was distinguished with high accuracy in the first half of chewing before the food was crushed. Conversely, the detection performance of food items other than the gum was lower in the latter half of the process, perhaps because the food was already crushed and spread in the mouth and therefore chewed as a whole. The data representing the correct answer could also be ambiguous because the labels for the correct answers were assigned to chewing positions on the basis of self-reporting by the subjects. Therefore, methods of evaluating chewing positions in the latter half of the chewing task must be further examined.

## 5. Conclusions

The present study investigated a method using eating sounds recorded from two channels of skin-contact microphones worn under the ear to efficiently and automatically detect left chewing, right chewing, front biting, and swallowing. The proposed method improved the MAPE from 33.6% for the conventional LSTM-CTC method to 19.5% by training the hybrid CTC/attention model, which can learn contexts with a 10-fold increase in data augmentation. In particular, it was found that the MAPE of left and right chewing improved through the use of the hybrid CTC/attention model. In particular, the detection performance of left and right chewing was high, with an F value of 0.85. It was also found that the detection performance of open food was almost identical to the identification of closed food, suggesting the possibility of the precise detection of food.

In the future, it may become possible to accomplish the exact detection of front biting and swallowing through the use of a neck-mounted or a close-talk microphone rather than an under-ear microphone. Furthermore, considering the utilization of other data augmentation approach, utilizing the speech recognition models such as the transformer and conformer model and development of applications to visualize eating behavior for health care will be explored.

## Figures and Tables

**Figure 1 sensors-21-03378-f001:**
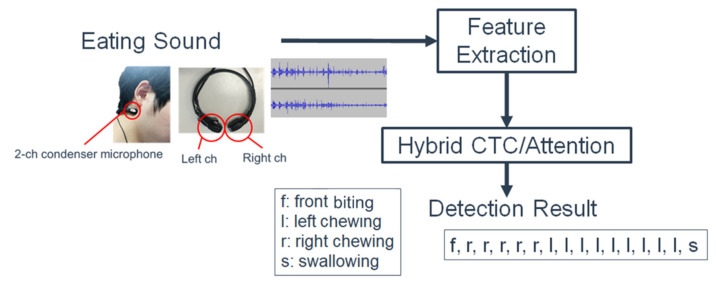
The automatic detection system of left chewing, right chewing, front biting, and swallowing.

**Figure 2 sensors-21-03378-f002:**
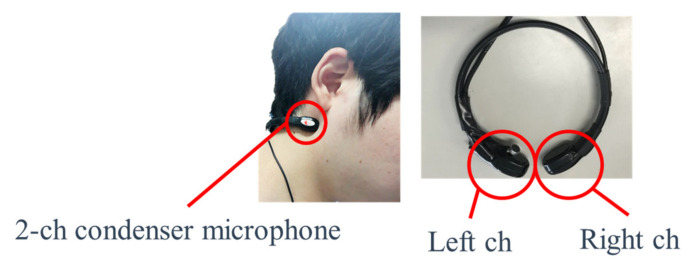
The installation of the microphone and the microphone unit.

**Figure 3 sensors-21-03378-f003:**
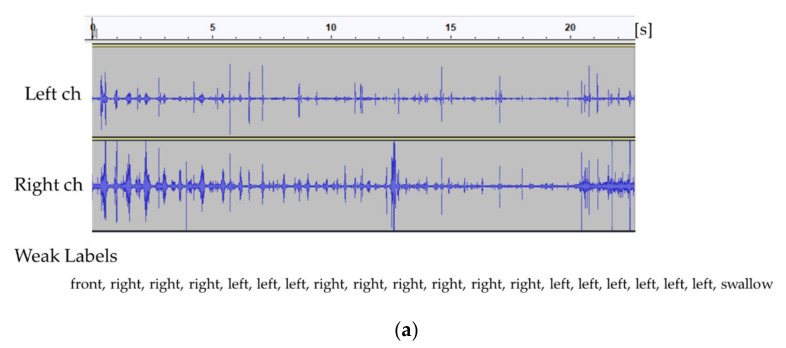
Recorded eating sounds (one cracker) and their corresponding logs for each subject (**a**–**d**).

**Figure 4 sensors-21-03378-f004:**
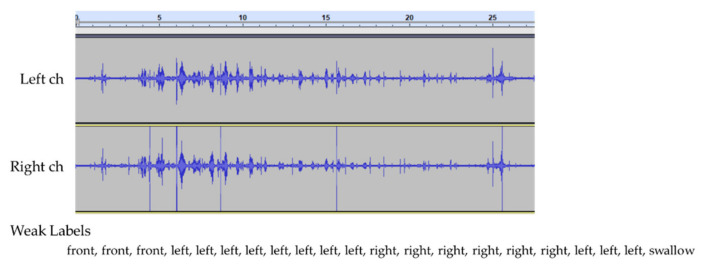
Sample recorded eating sound (two-channel) and event sequence (weak labels) for one cracker.

**Figure 5 sensors-21-03378-f005:**
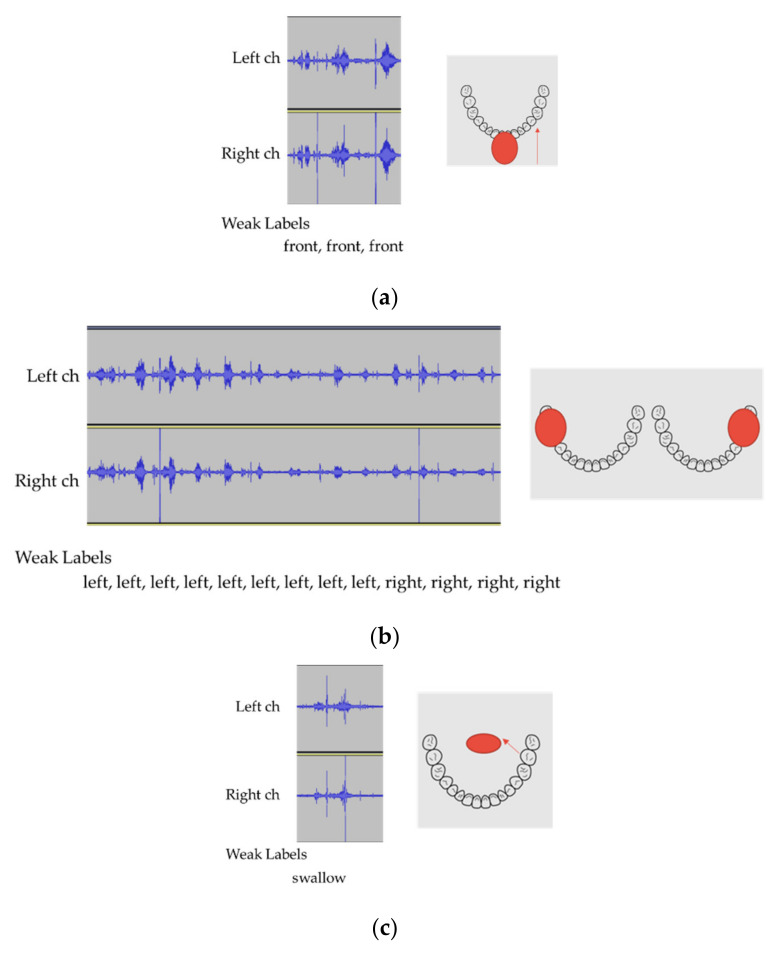
Motion for eating one cracker (**a**) front biting, (**b**) left and right chewing, and (**c**) swallowing.

**Figure 6 sensors-21-03378-f006:**
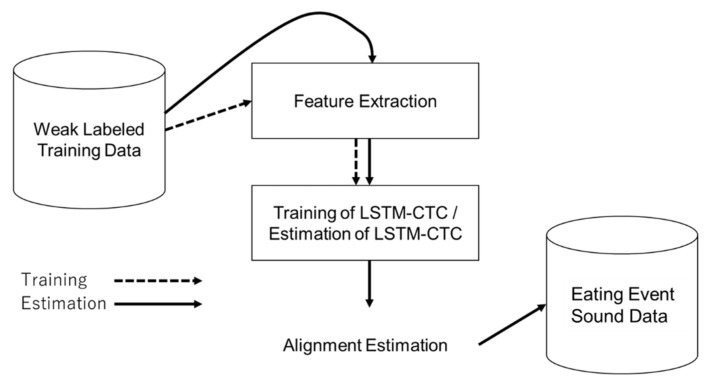
Division of event sound data using weak labeled data.

**Figure 7 sensors-21-03378-f007:**
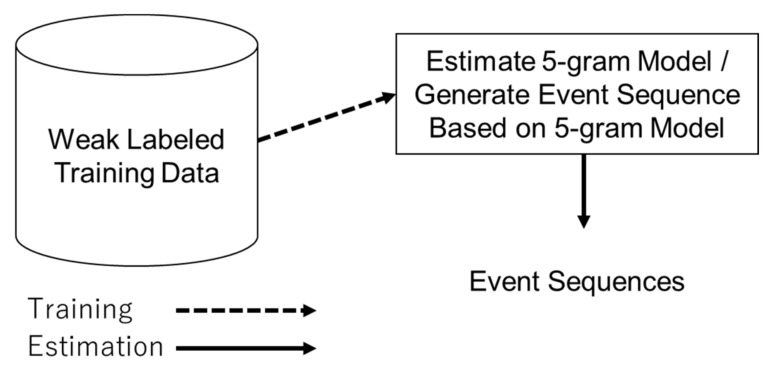
The 5-gram based event sequence augmentation.

**Figure 8 sensors-21-03378-f008:**
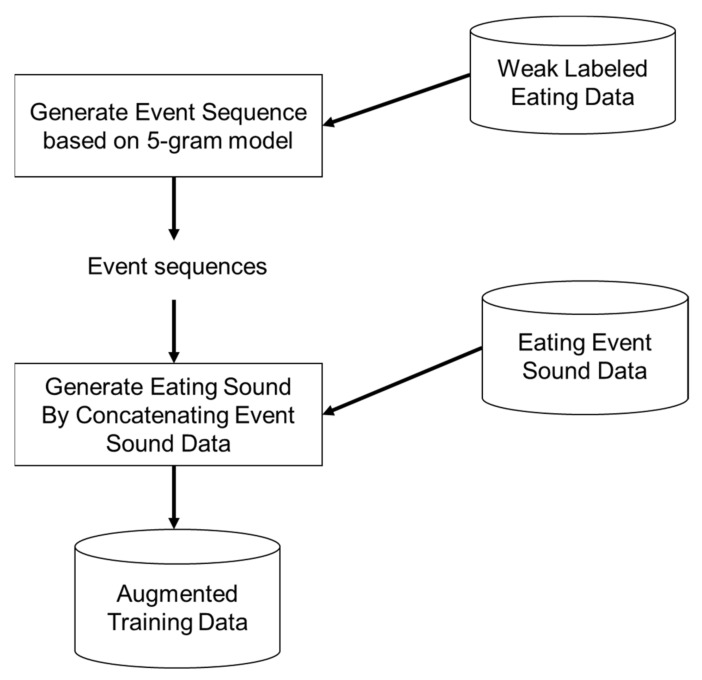
Data augmentation by concatenating eating event sound data.

**Figure 9 sensors-21-03378-f009:**
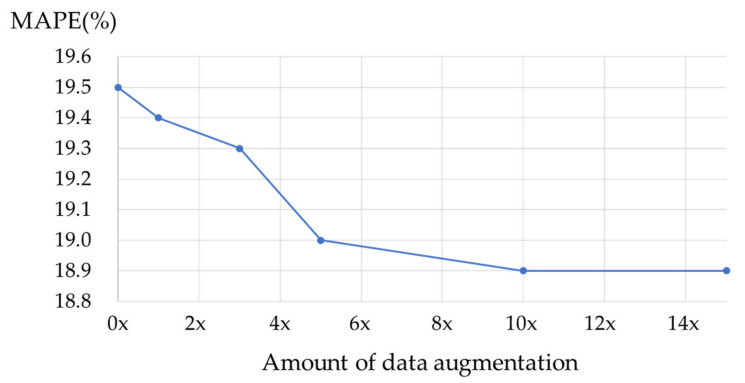
The relationship between the amount of data augmentation and overall detection performance (closed food and hybrid CTC/attention model): Five-class detection (left chewing, right chewing, front biting, swallowing, and others).

**Figure 10 sensors-21-03378-f010:**
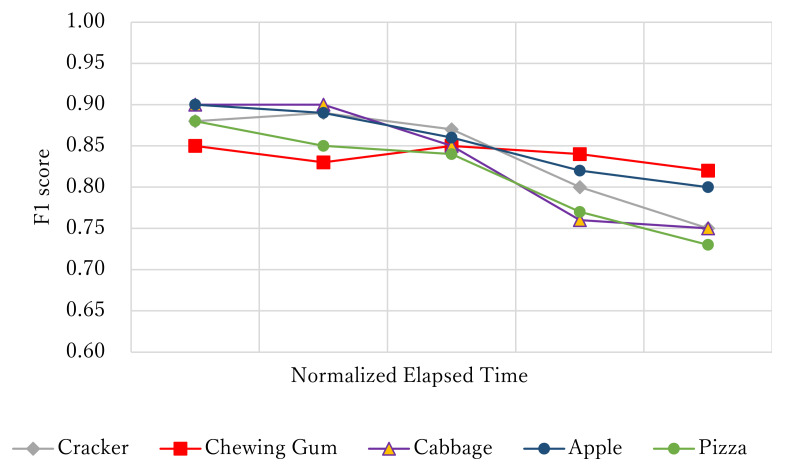
Relationship between normalized elapsed time and F1 score (open food, hybrid CTC/attention model, and 10× data augmentation): Five-class detection (left chewing, right chewing, front biting, swallowing, and others).

**Table 1 sensors-21-03378-t001:** The overall detection performance of the model (closed food): Five-class detection (left chewing, right chewing, front biting, swallowing, and others).

Model	MAPE (%)
Attention	46.3
CTC	33.6
Hybrid CTC/attention	**19.5**

**Table 2 sensors-21-03378-t002:** Per-event detection performance by model (closed food): Five-class detection (left chewing, right chewing, front biting, swallowing, and others).

	CTC			Hybrid CTC/Attention		
Event	Recall	Precision	F1 Score	Recall	Precision	F1 Score
Left chewing	0.74	0.83	0.76	0.83	0.87	0.85
Right chewing	0.76	0.82	0.78	0.85	0.89	0.87
Front biting	0.34	0.66	0.45	0.44	0.67	0.53
Swallowing	0.80	0.40	0.60	0.80	0.70	0.75

**Table 3 sensors-21-03378-t003:** The overall detection performance by sound feature (closed food and hybrid CTC/attention model): Five-class detection (left chewing, right chewing, front biting, swallowing, and others).

Sound Feature	MAPE (%)
MFCC of left signal	81.4
MFCC of right signal	85.3
Concatenation of MFCC of left and right signals	46.6
MFCC of the sum of left and right signals	51.1
Proposed * (3-point shift)	19.8
Proposed * (7-point shift)	**19.5**
Proposed * (15-point shift)	19.9

* Proposed: MFCC of the sum of left and right signals + cross-correlation of left and right signals.

**Table 4 sensors-21-03378-t004:** N-gram and overall detection performance (closed food and hybrid CTC/attention model): Five-class detection (left chewing, right chewing, front biting, swallowing, and others).

Model	MAPE (%)
Random	21.5
2-gram	20.0
5-gram	**18.9**
7-gram	**18.9**

**Table 5 sensors-21-03378-t005:** Per-event detection performance by data augmentation (closed food and hybrid CTC/attention model): Five-class detection (left chewing, right chewing, front biting, swallowing, and others).

Event	Baseline			Augmented (10×, 5-gram)		
	Recall	Precision	F1 Score	Recall	Precision	F1 Score
**Left Chewing**	0.81	0.87	0.84	**0.84**	**0.87**	**0.85**
**Right Chewing**	0.82	0.86	0.84	**0.82**	**0.88**	**0.85**
**Front Biting**	0.47	0.70	0.58	**0.56**	**0.80**	**0.68**
**Swallowing**	0.80	0.63	0.77	**0.86**	**0.74**	**0.80**

**Table 6 sensors-21-03378-t006:** The detection performance of chewing side by food type (closed food, hybrid CTC/attention model, and 10× data augmentation).

Food Type	Three-Class Detection *			Five-Class Detection **		
	Recall	Precision	F1 Score	Recall	Precision	F1 Score
**Gum**	0.92	0.93	0.92	0.83	0.88	0.85
**Cracker**	0.91	0.95	0.93	0.81	0.89	0.85
**Cabbage**	0.93	0.95	0.94	0.82	0.88	0.84

* Three-class detection: chewing, swallowing, and others. ** Five-class detection: left chewing, right chewing, front biting, swallowing, and others.

**Table 7 sensors-21-03378-t007:** Detection performance by food type (open food, hybrid CTC/attention model, and 10× data augmentation).

Food Type	Three-Class Selection *			Five-Class Detection **		
	Recall	Precision	F1 Score	Recall	Precision	F1 Score
**Apple**	0.89	0.85	0.92	0.82	0.89	0.86
**Pizza**	0.86	0.90	0.88	0.79	0.85	0.82

* Three-class detection: chewing, swallowing, and others. ** Five-class detection: left chewing, right chewing, front biting, swallowing, and others.

## Data Availability

Not applicable.

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
