# Peer review of "Automatic Detection of Chewing and Swallowing†"

_sensors, 2021, doi:10.3390/s21103378_

Round 1

Reviewer 1 Report

Interesting paper, but I have some concerns:

  • Which is the objective or application of detecting the 5 classes? Also, as the authors mention, it is clear that “the recording at the left and right position below the ear was specialized for the detection of left and right chewing and the amount of training data was insufficient compared to left and right chewing.” So, it seems there is an important bias.
  • The improvements with data augmentation are small. May be some alternatives could be explored
  • It is difficult to compare this work with others.
  • Why there is no work dedicated to food type detection, considering the database collected?

The values in Table 4 and Figure 8 are not clear. It seems 19.1 MAPE without data augmentation… then Table 4 is not clear

Minor typos:

Abstract:  maintenance maintaining -> maintenance

Page 3: CTC) -> CTC

Page 4: The present study reports that the hybrid model improved the CER to 8.4%.-> This text is repeated.

Author Response

Thank you for the thoughtful and constructive feedback you provided regarding our manuscript.

Reviewer 2 Report

In this paper the authors propose a system to automatically detect different sides of mouth chewing and swallowing of food using recorded sounds. They claim that their proposed study overcomes obstacles imposed in other studies such as the simple detection of the number of times and approximation of eating behaviors that they have not yet been able to attain information that can be used to analyze eating behaviors. The current study examines the automatic detection of left / right chewing, front biting, and swallowing using a hybrid CTC/attention model with 2ch microphones under the ear and weak labeled data for training to detect the balance of chewing and swallowing. Additionaly N-gram-based data augmentation was performed using weak labeled data to overcome the problem of reduced data as well as the problem of the correct labeling.

Reviewer's Comments
=======================
Generally, the paper is well written with monor grammatical / syntactical errors with good use of the English language.
However, some specific comments are given:
- In the second sction (Materials and Methods), subsection 2.1.1 the authors should include more state-of-the-art references in end-to-endspeech recognition.
- How did the authors select the position of the microphones? Have they tried any other positions on the face (like on top of the masseter muscle) for example? The authors should justify their choice of the recording site.
- In Figure 3 and 4 the authors should show the different classes of events (left-right-front-swallow) marked on the sound recordings. In this way the reader could see visually the different types of event sound recordings.
- Furthermore, they should describe in more detail the different morphology of the sounds that are recorded by the two microphones, depending on the side of the mouth that the subject chews (or swallows).
- The double threshold method (reference 18) should be more thoroughly presented and described in more detail.
- The paper lacks a discussion section where the experimental results should be evaluated and compared to similar recognition systems that also deal with the same kind of sound recognition problem. The authors have not succeded in showing in a clear manner the innovation of their method or how/why is their proposal is better than existing ones from the literature. 

Author Response

(The authors gave the same response as above.)

Round 2

Reviewer 2 Report

The authors have taken all comments into account, and have modified their submission accordingly. Therefore it is my opinion that their work can be accepted for publication.